# Open Urban Space Regeneration Strategies Based on Urban Welfare: A Project and Experiment in the San Lorenzo District in Rome, Italy

**Carmela Mariano** [1,*] , **Ignacio Gràvalos Lacambra** [2] **and Patrizia Di Monte** [3]

1   Department of Planning, Design, Architecture Technology, Sapienza University of Rome, 00196 Rome, Italy
2   Escuela de Arquitectura y Tecnologia, Universidad San Jorge, 50830 Zaragoza, Spain
3   Technical Office Urban Regeneration Estonoesunsolar, Project Partner Horizon 2020 gE.CO Living Lab, Arquitectos Grávalos Di Monte, 50001 Zaragoza, Spain
*   Correspondence: carmela.mariano@uniroma1.it

**Abstract:** The current socio-economic dynamics and the consequences induced by the pandemic emergency have generated a reflection on the need to recover the dimension of proximity and to share resources, spaces, infrastructures, and experiences. This solicits a remodelling of the system of public open spaces, based on a resilient, adaptive model; multifunctional and linked to the temporality of the functions that spaces can accommodate. The paper deals with the issues of planning and design of public open spaces around the needs of proximity and welfare. This is achieved through collecting and systematizing state of the art concepts on the role of public space within the urban structure of the city, and the formulation of guidelines for design, deduced from an empirical application conducted on a pilot district in the city of Rome. The paper aims to suggest to policymakers and planners a new approach and a path for future research and practice in the planning and design of more sustainable and inclusive green areas and public spaces, meeting the diverse needs of citizens. We undertook this objective through the experimental application of an intervention methodology on the public space system of the San Lorenzo neighbourhood in Rome.

**Keywords:** public space; sharing and inclusiveness; quality; urban regeneration; temporary uses

## 1. Introduction

The contemporary city is considered a place of density and physical, functional, social, and symbolic complexity [1]. In the early 2000s, it became a place where environmental and social crises intertwined; conditions exacerbated in recent years by the difficulties generated by the pandemic emergency [2]. Indeed, the pandemic has led to a condition of social vulnerability and has highlighted the condition of marginality in places far from the centre, to which must be added the inequalities related to income, education, work, proximity, and quality of services [3].

This new scenario has generated an increased awareness of the scarcity of environmental resources, combined with growing demands for safety, health and education, technological progress, and changing rules of social interaction [4]. These factors have undoubtedly caused effects on urban transformation processes anchored to the assumption of growth; cancelling, or at least considerably reducing growth's driving role [5].

In a few decades, we have witnessed a mutation of the hierarchies and urban balances that had been built over time, and, alongside the potential for development, lines of division and tension have emerged that have contributed to the emergence of increasingly explosive inequalities [6,7]: social segments excluded from the labour market and the enjoyment of goods, the privatization of public space, inequalities in the system of collective mobility, the new poverty of non-integrated immigrant realities, the issue of social housing and its segregation, security, and many others [8,9].

In this new urban dimension, public space has once again become the site of an increasingly transversal and difficult-to-govern conflict [10], which has produced a disaffection of the individual towards urban spaces [11,12] «to take refuge in the extraterritoriality of electronic networks (...) And so public space is increasingly emptying itself of public issues. It is incapable of fulfilling its past role as a place of encounter and debate, of private suffering and public issues» [13].

The current socio-economic dynamics and the consequences induced by the pandemic emergency have generated a reflection on the need to recover the dimension of proximity [14] and to share resources, spaces, infrastructures, and experiences. This reflection is also confirmed by the guidelines of the European Commission for the new urban agenda that identifies the perspective of the human-centered city [15], in which the role of communities and institutions «as city makers, co-creators of their evolving urban development and actors of innovation» [16] is central.

Numerous scientific types of research have highlighted the role of green areas and public open spaces on the health and well-being of citizens [17], as places of aggregation and relationships that stimulate social cohesion, reduce crime rates [18], promote the use of sustainable mobility and pollution reduction [19], and contribute to the pursuit of urban welfare [20,21].

Government agencies and organizations identify the improvement of human health and well-being as a priority urban goal and have increasingly promoted the creation of safe, healthy, inclusive, and sustainable urban environments in cities, particularly emphasizing the importance of building networks of green spaces and well-connected, well-distributed, and accessible public spaces to improve physical and mental health, urban livability, and to increase resilience to environmental hazards [22,23].

The lockdown experience induced by the most complex phases of the pandemic emergency in the years 2020–2021, with its restrictions on urban mobility, has further underlined the importance of designing our cities with sustainable design principles, encouraging people to walk, cycle and enjoy public open spaces [24]. The pandemic has changed established social behaviours, especially in urban areas where the need for outdoor gathering spaces is felt. For this reason, the issue of proximity, linked to urban everyday life and to neighbourhoods and their ability to respond to citizens' needs, calls for a remodelling of the system of public open spaces based on a resilient, adaptive model [25]; multifunctional and linked to the temporality of the functions that spaces can accommodate [26].

In this context, the paper addresses the issues of planning and design of public open space around the needs of proximity and welfare, through a work of collection and systematization of the state of the art concepts on the role of public space within the urban structure of the city, and the formulation of guidelines for design deduced from an empirical application conducted on a pilot neighbourhood in the city of Rome.

The paper aims to suggest to policymakers and planners a new approach and pathway for future research and practice in the planning and design of more sustainable and inclusive green areas and public spaces that meet the diverse needs of citizens.

## 2. Literature Review

The hierarchy of needs elaborated by Maslow (1968) [27] is a psychological theory that identifies five categories that determine an individual's behaviour. These are physiological needs; security, belonging, love, esteem, and self-fulfillment.

The different dimensions of public space—in its morphological, perceptual, visual, relational, functional, and temporal meanings [28,29]—are closely related to the hierarchy of needs and convey the complexity of this urban infrastructure and the need for 'community building' [2]. This is only possible by creating an environment suitable for the production of stimuli, social relations, and a multiplicity of events, i.e., a system of proximity [30,31] sufficiently diversified and balanced between the functional and relational dimensions of public spaces.

The debate on the need for a balance between these two dimensions dates back to the 1960s, starting with the theories of Jane Jacobs (1961) [32], who identified the performance of functions as the indicator of urban quality of public space. Following this, the theories of Lofland (1998) [33] speak of public space as the physical and morphological dimensions of public space, and the public realm as the relational dimension involving the social interaction between inhabitants. Montgomery (1998) [34] identifies the success factor of public space as the correct combination of activities performed, and the meaning and quality of the physical environment. In more recent years, Jan Gehl argues that the quality of the public space arises from the everyday practices and behaviours that take place in the street, with 'the presence of people, the production of events, activities, stimuli, solicitations' [35,36].

In subsequent years, quality indicators for public spaces have been redefined according to changing socio-economic dynamics. According to Carmona [28,29], key factors are related to accessibility, perceptual and visual comfort, and usability of spaces in social terms. Richard Sennet (2019) [37] also states that urban quality depends on the interaction, more or less dialogic or conflictual, between the components of the built environment (the Ville) and that of the lived environment (the Cité), i.e., between the functional proximity embodied by the Villé and the relational proximity realized in the Cité.

In this context, the search for a renewed relationship between urbs and civitas calls for the contribution of the town planning discipline to adapt the design of public spaces to new social practices and new needs. This prefigures a complex design capacity in which space and society, the physical dimension, listening to demands, overall visions, selective actions, planning, programming, large and small scales, long and short timescales, strategies, rules, and projects constantly interact [38]. All this, to restore 'depth' and 'density' [39] not only to the physical and morphological dimension of public space, but above all to the relational dimension and the set of social interactions between inhabitants; a dimension that is not visible and representable but fragmented and changeable [33].

The transformation dynamics of the contemporary city in recent decades, which are more changeable and therefore more uncertain, make it difficult to foresee and anticipate [40,41], highlighting the need to identify a new approach to the design of public spaces: more strategic and better able to adapt to unforeseen situations and events. An approach that stimulates the adaptive capacity of cities, through innovative experimentation and a series of actions adapted to the speed of urban transformations [42,43] and capable of innovating planning and design tools and procedures, is also called for by the guidelines of goal 11.7 of the Sustainable Development Agenda: "By 2030, provide universal access to safe, inclusive, and accessible green and public spaces, in particular for women and children, older persons and persons with disabilities".

Hence, the design intervention in the system of collective open spaces, residual areas, and disused areas, in both urban contexts of the historical city and the most marginal areas of the modern periphery, is increasingly confronted with the need for an integrated approach to urban complexity [44]. This can make a sustainable turn, in a polysemous sense, in urban regeneration strategies and the construction of the 'public city'. This may be with particular reference to the relations between urban well-being and the quality of spaces, between temporary uses, between a sense of belonging to places and identity, between new technologies, between the accessibility and use of spaces, between participatory processes, and between the effectiveness of local/urban public policies and social cohesion.

The design intervention in the system of public spaces should provide temporary, non-specialized, multifunctional, hybrid, and reversible functions [45]; favouring a transitional and incremental approach of regeneration [46] as a design device that is capable of accompanying the consolidation of lasting uses over time and the identity and sense of belonging of the community [47,48].

A perspective of action, in which the designer assumes the role of placemaker [49] engaging in concrete processes of constructing new economies, new possibilities for living, and new forms of sociability. This also permeates the action of the New European Bauhaus

Programme [50], relaunching the challenge of combining cultural and social innovation in the design of public spaces with the principles of sustainability and spatial justice [51–53], promoting the involvement of stakeholders and the satisfaction of community needs [54,55].

## 3. Materials and Methods

### 3.1. Materials

The paper proposes the experimental application of an intervention methodology on the system of public spaces in a neighbourhood of the historical city of Rome, deduced through an inductive method from a conceptualization of theoretical–methodological and operational references for the design of public spaces in the context of the contemporary city. This was started from the research activities carried out by the PDTA Department of the Sapienza University of Rome, in collaboration with the Escuela de Arquitectura y Tecnologia, Universidad San Jorge, and with the Project Partner of the Horizon 2020 project gE.CO Living Lab. This research activity involved the study and critical evaluation of the results of several design experiments conducted by the Gràvalosdimonte design studio in Zaragoza. This made it possible to elaborate a toolkit of actions and guidelines for intervention in systems of public spaces in the contemporary city, and constitutes one of the most interesting results of the Horizon 2020 gE.CO Living Lab project. The toolkit was then tested on a pilot neighbourhood in the historic city of Rome.

In particular, in the preliminary research work on design experiences, the experimental programme Estonoesunsolar in Zaragoza and the Supermanzana project in Barcelona were analyzed. These combine the theme of the interscalarity of intervention (territorial, urban and local scales) with the theme of flexibility of functions concerning local instances and contingencies, with particular reference to the topical need for urban health, the dialectic between public and private space from the perspective of the notion of the common good, bottom-up design for the construction of neighbourhood spaces, and plurality in decision-making [56–58].

In addition, the research and experimentation activity also made use of the intervention guidelines contained in the Temporary Use Toolkit developed by arch. Di Monte as Project partner of the Horizon 2020 gE.CO Living Lab project [59]. This identified tactical intervention methodologies from the perspective of proximity [60], emphasizing the criteria of safety, inclusion, connectivity, accessibility, and sustainability.

The document establishes a 6-step methodological tool for the design of empty spaces in contemporary city contexts and their reactivation in a new life cycle. The methodology includes the steps:

- Offer: which corresponds to a mapping of empty spaces including properties, state of maintenance, present uses, and functions;
- Demand: which corresponds to reconnaissance and evaluation of local demand, and bottom-up initiatives about the transformability of spaces with the elaboration of design concepts;
- Assignment and funds: corresponding to the stages of involvement of stakeholders (public/private and local associations) in the drafting of the business plan for the identification of economic and social benefits;
- Implementation: corresponding to the project authorization phase and the identification of local action groups for project implementation;
- Assessment: corresponding to the monitoring, management, and reporting phase of the project results.

This document implements the guidelines of the United Nations Agency UN-Habitat's key messages on COVID-19 and public space (2020). In order to provide an effective response to the critical issues arising from COVID-19 in the urban context, these guidelines identify three main studies needed to quantify the new demand for the use of public space and to prepare short-, medium- and long-term supply strategies:

- conduct spatial analyses at the urban level, aimed at mapping resources and highlighting resource and vulnerability points;

- understanding all the functions to which open spaces in the city respond and the levels of accessibility;
- rethinking the provision of public space in terms of temporariness, flexibility, and multi-functionality.

The Experimental Programme 'Estonoesunsolar', promoted by the Sociedad Municipal Zaragoza Vivienda, is an experimental program for the temporary use of abandoned lots in the city of Zaragoza. Between 2009 and 2010, this saw the realization of 32 interventions, reconverting 60,000 square meters of disused spaces into new public spaces [58] with multiple uses and functions representing a concrete response to neighbourhood demand (Figure 1).

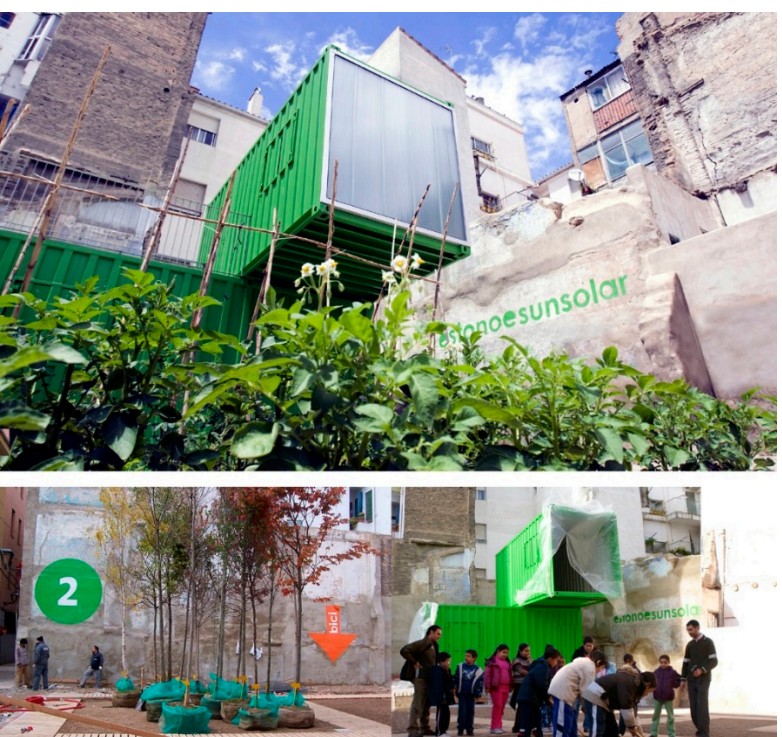

**Figure 1.** Zaragoza waterfront regeneration intervention—estonoesunsolar. The number 2 corresponds to the progressive numbering of the 32 regenerated spaces in the city and the arrow refers to the bicycle parking space.

'Estonoesunsolar' was an innovative response to the large number of 'meanwhile spaces' [61] that have shaped the urban landscapes of the crisis. It focused on the social dimension of public space as a place of exchange and interaction, and as an expressive context for multiple conflicts and, thus, urban learning [32]. It was precisely this condition that succeeded in the conceptual conversion of physical space into place, understood in its anthropological sense. It contracted the distance between the concepts urbs and civitas [62] or, in a Lefebvrian's words, between the urban fabric and «the modus vivendi» [63] of urban society.

Each of the proposals investigated the meanings of the urban context, seeking to enhance elements of the collective memory of the place/ To some extent, this was constructing a space in which the affective bonds that constitute the "murmur of societies" [64] could be strengthened. In the context of an eroded public space, the intention was to create elements that would constitute small affective anchorages within the territory; spaces of emotional significance opposed to what Daniel Hiernaux calls "the slippery city" [65], through which one passes without leaving a trace (Figure 2).

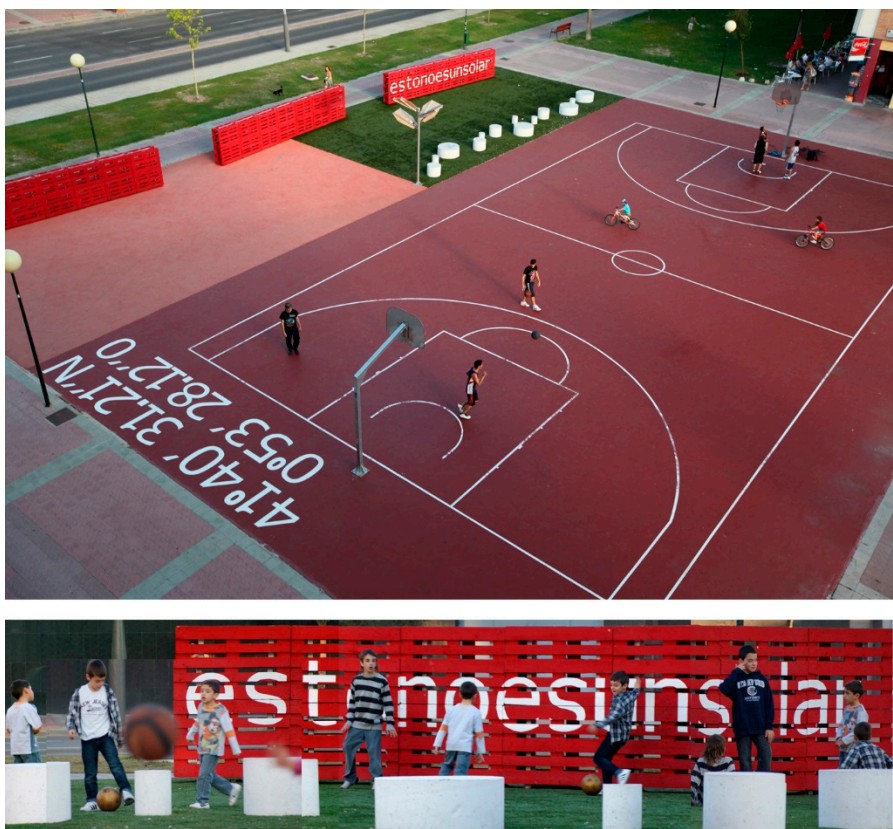

**Figure 2.** Zaragoza Estonoesunsolar Programme.

The pilot experience of the first 'functional superblock' was part of a workshop organized by the International University of Catalonia in 2016. Students from all the universities of architecture in Barcelona participated. The course aimed to test the new public spaces resulting from the implementation of Barcelona's new Urban Mobility Plan (2013–2018) in the Poble Nou district of Barcelona.

The 'superblock' is a model comprising 9 blocks of the city peripherally surrounded by major road and public transport networks. It has a characteristic of soft mobility: a marked predominance of pedestrian traffic. The four inner crossings of the superblock, previously intended for vehicular traffic, were to be converted into pedestrian spaces. This created a kind of micro-city within the city, where pedestrians were to have absolute priority.

The pilot experience, developed in experimental terms in the workshop, offered keys to understanding very simple and effective urban strategies. These were subsequently taken up and promoted by the Barcelona City Council in the post-pandemic emergency. They were promoted within the post-Covid mobility plan as provisional solutions, although they aimed to be incorporated within a more radical urban transformation strategic process.

Transforming traffic spaces into spaces for pedestrians cycling and public transport improves air quality, promotes physical activity, and curbs climate change. The new Soft Mobility Plan envisages a 270% increase in pedestrian travel, consequently, pedestrians will be able to use between 45% and 70% of public spaces.

Initially, the intervention addressed the issue of the legibility of space. The substantial change in mobility has entailed the rearrangement of roles in the new pedestrian–vehicle dialectic. For this, it was necessary to assign new reading codes to the collective imagination [66]. For this reason, one of the proposed objectives was to endow the space with a new identity making the new limits of the pedestrian zone legible and that would also convey the new environmental values of scarce and restricted circulation. To implement this, it was decided to use a recognizable icon, the 'panot', in outsized application to homogenize the space and create the pedestrian area (Figure 3).

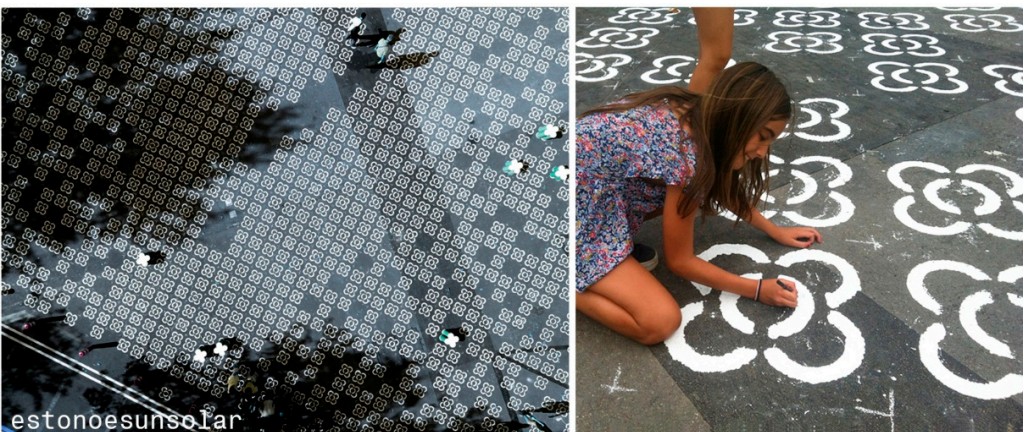

**Figure 3.** New public space of the 'supermanzana' in Poblenou.

*3.2. Methods*

In both case studies described, interventions were made within the already-built city through urban acupuncture [67,68]. Far from becoming self-referential interventions, this allowed a very direct relationship to be established between tactical action and urban planning.

From the outset, experimenting with the infinite possibilities of temporary uses, while maintaining the need to regulate these good practices within a legal framework, would enable them to become permitted interventions. In the case of the 'supermanzana', which was a pilot project, it subsequently became a replicable model in other urban contexts.

The reference to 'meanwhile spaces' stimulates and legitimizes the adoption of temporary solutions. In the cases represented, they are preludes to regulated, larger-scale urban transformation strategies. Both cases desire to rebalance and reprogram the existing city through experimental approaches that seek to adapt the city to contemporary contingencies.

Starting from the study of the process and project methodology used in these experiences and based on the results achieved, it was possible to elaborate a project and intervention methodology on the system of open and public spaces. This was articulated in three distinct phases that take up and update some of the methodological references of the Temporary Use Toolkit and the UN-Habitat 2020 Document.

1. **Offer**

    The first phase consists of:

- Analysis and mapping of the system of public open spaces (squares, streets, and green areas) and private spaces for public use in the district. This was built from a series of physical inspections, from the verification of territorial cartographies and orthophotos, and from thematic elaborations of the cognitive framework of the planning tools that concern the area under study;
- Mapping the offerings of neighbourhood services and activities present in the case study district, catalogued by functional areas, obtained through a census using Qgis software and open-data territorial information systems.

2. **Evaluation and Demand**

    The second phase aims to return an evaluation of the results of the analysis conducted in Phase 1 according to criteria linked to the proximity requirement of the open-space system. This phase consisted of:

- Evaluation of the level of accessibility of each space, the function present, and the system of connections (visual and physical) within the open-space system;
- Evaluation and synthesis of local demand, involving the main social associations operating in the area in the identification of flexible uses and functions of open spaces, with a view to networking and enhancing these.

### 3. Proposals

The third phase aims to identify some project scenarios consisting of:

- hypotheses for reuse and identification of flexible functions for the open-space system;
- identification of stakeholders, consultation and sharing with stakeholders, and drafting phases and timetable of interventions;
- project examples.

This methodology was subsequently tested on an experimental basis in the historical San Lorenzo district in Rome.

### 4. Case Study: San Lorenzo, from 'Island' to 'Inclusive' District

*4.1. Historical Background*

The San Lorenzo district arose between 1884 and 1888 and underwent several transformations and extensions during the 20th century, due to processes of building replacement of the original typologies and the reconstruction of bombed buildings in 1943 (Figure 4).

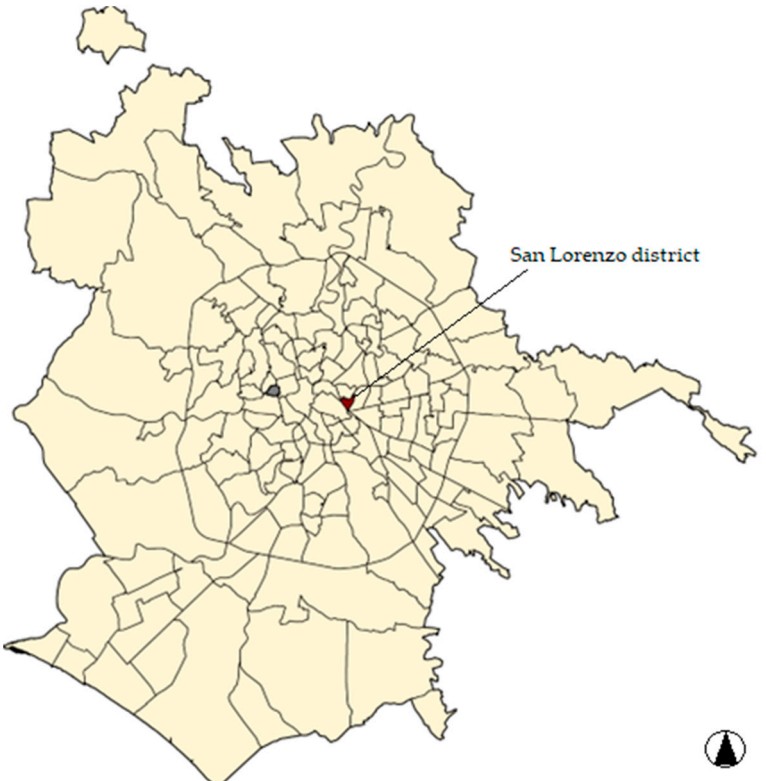

**Figure 4.** Location of the San Lorenzo district within the perimeter of the city of Rome.

At the beginning of the 19th century, the area occupied by the San Lorenzo district was completely undeveloped. Only the main ordering elements were present, such as the Basilica of San Lorenzo, the cemetery and the historical routes of the Tiburtina and Collatina [69]. Later came the construction of the railway and Termini station in the areas between the Walls and the Via Tiburtina, which abandoned its ancient route with the opening of a gap in the Walls and the opening of the Tiburtino square (IGM, 1872–1874, 1906, 1924). Particularly in the early years of the 20th century, with the construction of modestly sized factories, such as the Paszko-wsky (later to become Wurer beer) in 1902 and the Cerere mill and pasta factory in 1905, the district started to become a compact working-class core. This was the period of a social and cultural development parallel with the spread of a political ideology that is still prevalent in the district today. This manifests itself through a pronounced class consciousness and solidarity, reinforced by the San Lorenzo inhabitants' sense of belonging to the neighbourhood community. During

the Second World War, San Lorenzo was the first district of the city to be bombed and was the hardest hit. Even today, it is still possible to find buildings in the neighbourhood damaged by the 1943 bombing that have not yet been rebuilt. The end of the 20th century was the period of reconstruction, with the replacement, extension, and completion of existing buildings. [70] (Headquarters of the Faculty of Psychology in Via degli Apuli and Neuropsychiatry in Via dei Reti). Starting from the end of the 1960s, the neighbourhood began to undergo a gradual process of 'depersonalization'; these were the years in which several public works of great importance for the neighbourhood were carried out, such as the building of Child Neuropsychiatry in via dei Reti and via dei Piceni and the Tangenziale Est. The neighbourhood began to be populated by out-of-town university students and little remains of the original population [71].

### 4.2. Main Features of the District

The urban layout of this part of the city owes its current definition partially from references to historical structures and ordering (Via Tiburtina, Aurelian Walls), but above all to large 'enclosures' of post-unification formation. These grew up immediately outside the city walls to provide the city with superior facilities and services: The Polyclinic, the University City, the Verano cemetery, and the military and railway areas of Termini Station and the Freight Station, which surround the residential tissues of the district. This constitutes a sort of "island", connected to the rest of the city by a high-density urban-level road system, and by tunnels under the bundle of tracks (San Lorenzo Urban Regeneration Programme, 2019). This variety of 'urban isolation' has reinforced the sense of territorialism and community belonging of the inhabitants of San Lorenzo. The character of the San Lorenzo district, from its earliest historical stages, has been accustomed to change. It is evident that, from its inception, the urban layout has influenced the social, economic, and morphological character of the district. From the very beginning, the urban structure had an orthogonal matrix with the Labicane walls, the freight yard, the Verano cemetery, and the Via Tiburtina as its limits. The configuration of the district and its position outside the city walls contributed to isolating it from the rest of the urban fabric, making it a true village within the city [72,73].

Urban isolation is accompanied by social isolation. The pronounced social characterization of its inhabitants is induced primarily by the district's economic and professional conformation; the fact that the railway freight yard, the water reservoirs of the main aqueducts, the garbage depot, the tram depot, the workshops for the construction of tram vehicles, the central railway station, the Verano cemetery, and the construction sites all around the expanding city are located here. This meant that mainly labourers, bricklayers, railway workers, tram drivers, artisans, and garbage collectors, as well as those linked to cemetery activities, such as carpenters, welders, stonemasons, tinsmiths, construction workers, and marble workers were housed in the district.

The presence of the headquarters of La Sapienza University, Europe's largest university, has, over the years, progressively altered the social structure of the neighbourhood. As a result, an increasing number of out-of-town university students have been added to the resident population. This settlement pressure has led, over time, to a clear mutation of the district's socio-economic characteristics. The progressive decline of traditional craft activities has benefitted the proliferation of tertiary-commercial and public service activities. This has led to the phenomenon of gentrification, an increase in traffic volumes, rising rental costs, and a crisis for local crafts (Figure 5).

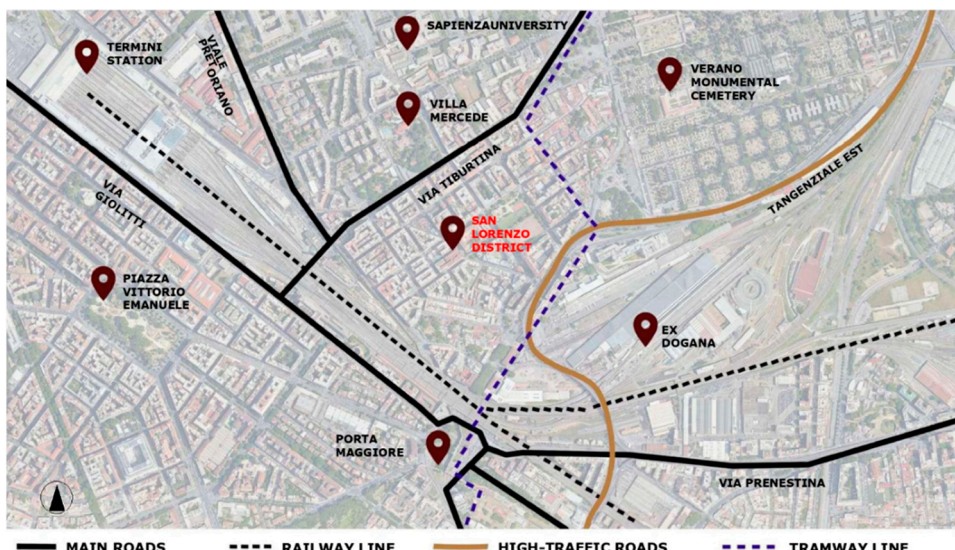

**Figure 5.** The urban structure of the district.

Moreover, this process of mutation and the urban dynamics that have affected the neighbourhood have highlighted the lack of services and amenities and neglected the neighbourhood's emerging characteristic; namely, the wealth of historical and monumental evidence to be found there.

For these reasons, the San Lorenzo district, which falls within the connotations of the Historic City in the General Urban Plan of the Municipality of Rome, [74] has been the subject of a redevelopment plan launched by the Municipal Administration in 2006, as an integral part of a broader and more articulated Urban Project called "San Lorenzo—Circonvallazione Interna—Vallo Ferroviario". It is closely connected to the definition of the mobility layout (New Inner Ring Road) and the railway areas [75]. The PRG also identifies several perimeters destined for type-B "Enhancement Areas" (PRG Sistemi e Regole, 2008), which "concern places that over time have not achieved or have lost the identity characteristics proper to the Historic City and are characterized by the presence of buildings and artefacts that are no longer used and can be reconverted to new uses or that present evident phenomenon of physical and functional degradation" (art. 43 paragraph 1 Implementation Technical Regulations).

Within this as-yet unfinished framework of a general strategy for the regeneration of the neighbourhood, there has been a lack of attention to the local system of collective open spaces, which constitute an identifying feature of the neighbourhood. In this sense, numerous initiatives have been launched and are planned for consultation and planning with the participation of citizens.

The experimental research activity considered the complexity of the neighbourhood's social structure and local demand, through the direct involvement of neighbourhood associations and citizens' committees rooted in the area. Several interviews were conducted with the associations' contact persons. Guided surveys were carried out to provide a picture of the resources, potential, and criticalities of open spaces and of local demand for their transformability, more generally.

## 5. Results and Discussion

### 5.1. Phase 1. Offer

The diagram depicted in Figure 6 represents one of the results of Phase 1. It offers and returns a mapping of the district's public and semi-public spaces (set about each other and graphically highlighted as voids) that made it possible to group the inner courtyards into four areas, delimited by the main road axes, each of which presents different morphological and functional characteristics (Table 1).

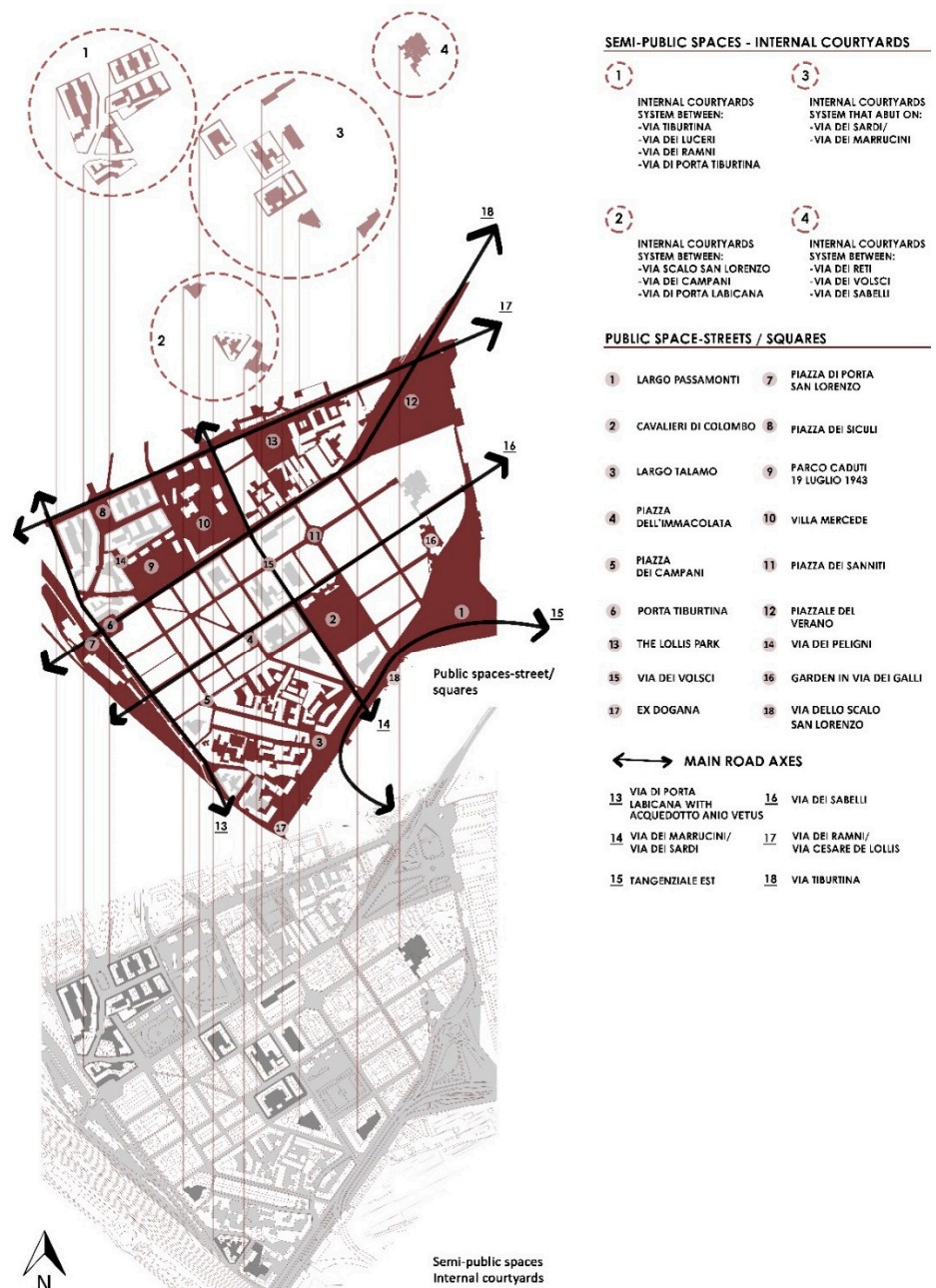

**Figure 6.** Mapping public and semi-public spaces in the San Lorenzo district (original scale 1:10,000).

The courtyards of group 1 are located between Via Tiburtina (18), Via dei Ramni/ Via Cesare de Lollis (17), and Via di Porta Labicana/Acquedotto Anio Vetus (13), and enjoy the proximity of important public spaces in close connection with each other: Porta Tiburtina (6), Piazza di Porta San Lorenzo (7), Piazza dei Siculi (8), Parco dei Caduti 19 luglio 1943 (9) and Villa Mercede (10). It follows that the re-functionalization of these semi-public spaces will have to be configured as a connective tissue of the social aggregation functions proper to the public spaces and the private spaces of the buildings.

The courtyards in group 2 are between Via dello Scalo San Lorenzo, Via dei Campani, and Via di Porta Labicana (13), and enjoy the proximity of Largo Talamo (3) and Piazza dei Campani (5).

In this case, the proximity of the Labicane Walls to Piazza dei Campani and the Largo Talamo tomb gives this complex of courts a strong identity connotation, from the point of view of historical memory.

**Table 1.** Phase 1. Relations of courtyards with road axes and public spaces.

| | Relations of Courtyards with Road Axes and Public Spaces | | | |
|---|---|---|---|---|
| **Public Spaces** | **Courts Group 1** | **Courts Group 2** | **Courts Group 3** | **Courts Group 4** |
| 1. Largo Passamonti | | | • | |
| 2. Cavalieri di Colombo | | | • | |
| 3. Largo Talamo | | • | | |
| 4. Piazza dell'Immacolata | | | • | |
| 5. Piazza dei Campani | | • | | |
| 6. Porta Tiburtina | • | | | |
| 7. Piazza di Porta San Lorenzo | • | | | |
| 8. Piazza dei Siculi | • | | | |
| 9. Parco Caduti 19 luglio 1943 | • | | | |
| 10. Villa Mercede | • | | • | |
| 11. Piazza dei Sanniti | | | • | |
| 12. Piazzale del Verano | | | | • |
| **Main Road Axes** | | | | |
| 13. Via di Porta Labicana/Acquedotto Anio Vetus | • | • | • | |
| 14. Via dei Marruccini/Via dei Sardi | | | • | |
| 15. Tangenziale est | | | • | |
| 16. Via dei Sabelli | | | | • |
| 17. Via dei Ramni/Via Cesare de Lollis | • | | | |
| 18. Via Tiburtina | • | | • | |

The courtyards of group 3 are located at the intersection of Via dei Marrucini/Via dei Sardi (14) and Via dei Sabelli (16), and are also related to via Tiburtina (18) and Tangenziale Est road (15). This group enjoys the proximity of the Piazza dell'Immacolata (4) Cavalieri di Colombo (2) and Largo Passamonti (1), functions that connote that the area has a strong sporting vocation. This suggests the inclusion of sports-related functions for the re-functionalization of the semi-public spaces present here.

The courtyards in group 4 between Via dei Reti, Via dei Volsci, and Via dei Sabelli (16) enjoy the proximity of Piazzale del Verano (12) and the Sapienza University, a circumstance that prefigures these spaces as a meeting places for residents and students.

The mapping of neighbourhood services and activities (Figure 7), relative to Phase 1 Offer, returns the consistency of the territorial endowments of the San Lorenzo district, concerning five prevailing areas of functions and activities:

- Culture;
- Free time;
- Work;
- Commercial activities;
- Transport.

This analysis was conducted using Q-GIS software from the Opendata of the GIS Cartographic Portal of the Metropolitan City of Rome, the Territorial Information System of Rome Capital, and the Romamobilita.it dataset of the Municipality of Rome Capital [76–78].

With specific reference to 'Culture', the diagram shows that the prevailing activities are those related to education and history, with the discrete presence of the social center and libraries.

On the other hand, for 'Free time', the district is endowed with numerous activities related to catering, including bars and restaurants; but is poor in cinemas, theatres, and public parks.

Where the item 'Work' is concerned, the diagram shows that the area is not characterized as an office pole, due to the scarce presence of offices.

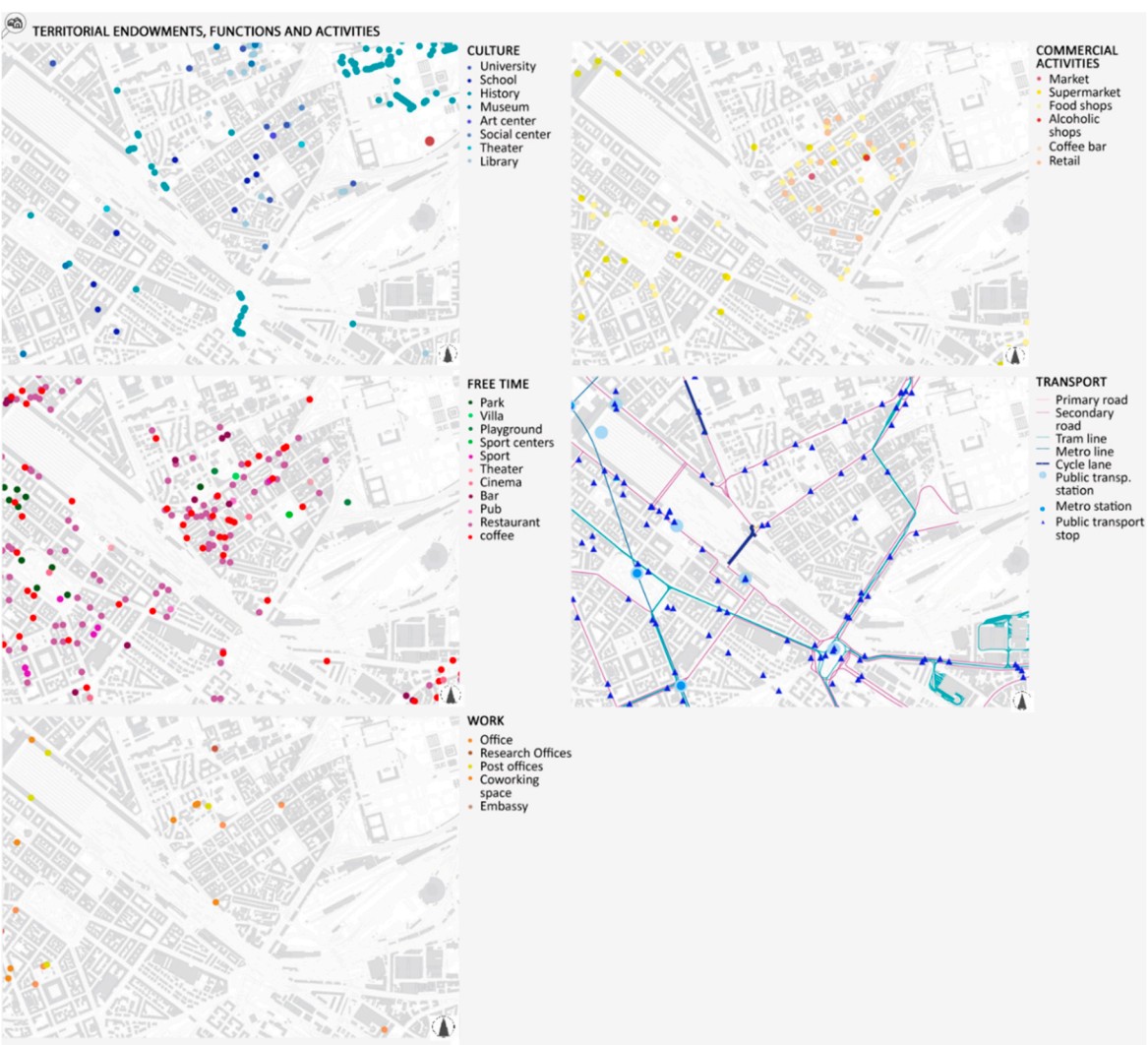

**Figure 7.** Mapping the supply of neighbourhood services and activities in the San Lorenzo district (original scale 1:20,000).

With respect to 'Commercial activities', on the other hand, it is evident that the predominant business activities are those related to the sale of food, with a modest presence of retail activities.

As far as 'Transport' is concerned, the San Lorenzo district is very accessible thanks to the massive presence of public transport, by rail and by road. In general, it has a good primary and secondary road network. It is noted that the cycle track routes are poorly laid out and disconnected, thus not guaranteeing optimal cycle accessibility.

Given the proximity of Sapienza University, it can be said that the San Lorenzo district has, over the years, modified the configuration of its territorial endowments to make them adhere to the needs and requirements of the students who live there. In some cases, this has neglected other categories of citizens and, thus, favouring the aforementioned processes of gentrification.

### 5.2. Phase 2. Evaluation and Demand

The diagram in Figure 8 shows some of the results of Phase 2: Evaluation and Demand. It proposes the layout of the public and semi-public spaces in the neighbourhood (existing and planned), numbered in the plan from 1 to 23, according to the four prevailing aspects envisaged by the project: planned movements, visual connections, temporary functions, and accessibility (Table 2).

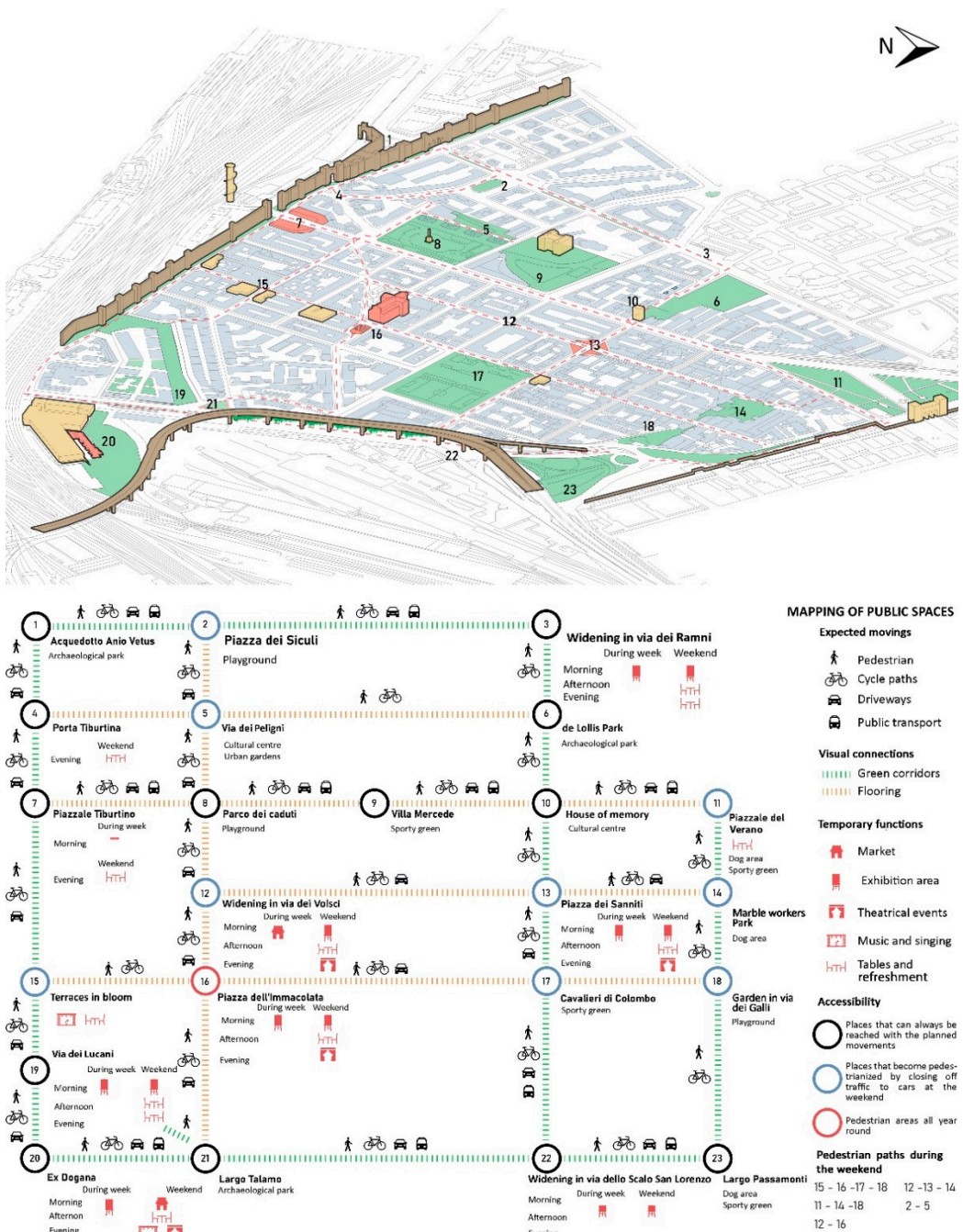

**Figure 8.** Levels of accessibility, functions, and connections of open spaces in the San Lorenzo district.

Firstly, the proposal identifies the planned modes of travel from one space to another; therefore, the diagram are indicates with dedicated symbols the stretches that can be travelled on foot, by bicycle, by car, or by public transport, envisaged in the project.

At the same time, the visual connections of these routes are highlighted, indicating the presence of green corridors or paving.

Furthermore, within the mesh generated by the diagram, the complex of the neighbourhood's courtyards is symbolically restored. This defines a series of temporary functions that vary according to the time of day (morning, afternoon, evening) and time during the week (midweek, weekend). These functions are: market, exhibition area, theatre, music and singing, tables, and refreshments.

In line with the temporality of the envisaged functions, the project proposal also envisages a modulation in terms of accessibility, indicated on the diagram with; black circles, places that can always be reached with the movements envisaged by the project, and shown in the diagram in the previously described modes of movement; blue, places for which the project envisages closure to vehicular traffic at weekends; red, places for which the project envisages only pedestrian use all year round.

The proposal returns a holistic vision of urban regeneration that emphasizes the importance of a systemic approach in both the analysis and design phases.

**Table 2.** Phase 2. Modes of travel and accessibility level.

| Planned Displacements between Public Spaces (The Numbers Refer to the Public Spaces Shown in Figure 8) | |
|---|---|
| On foot | 1–2; 2–3; 1–4; 2–5; 3–6; 4–5; 4–7; 5–8; 5–6; 6–10; 7–8; 7–15; 8–9; 8–12; 9–10; 10–11; 10–13; 11–14; 12–13; 12–16; 13–14; 13–17; 14–18; 15–16; 15–19; 16–17; 16–21; 17–18; 17–22; 18–23; 19–20; 20–21; 21–22; 22–23 |
| By bicycle | 1–2; 2–3; 1–4; 2–5; 3–6; 4–7; 5–8; 5–6; 6–10; 7–8; 7–15; 8–9; 8–12; 9–10; 10–11; 10–13; 11–14; 12–13; 12–16; 13–14; 13–17; 14–18; 15–16; 15–19; 16–17; 16–21; 17–18; 17–22; 18–23; 19–20; 20–21; 21–22; 22–23 |
| By car | 1–2; 2–3; 1–4; 2–5; 4–7; 5–8; 7–8; 7–15; 8–9; 8–12; 9–10; 10–11; 12–13; 12–16; 13–14; 13–17; 15–19; 16–17; 16–21; 17–22; 19–20; 20–21; 21–22; 22–23 |
| By public transport | 1–2; 2–3; 7–8; 8–9; 9–10; 10–11; 17–22; 20–21; 21–22; 22–23 |
| Accessibility Level | |
| All modes of travel | 1, 3, 4, 6, 7, 8, 9, 10, 19, 20, 21, 22, 23 |
| Pedestrian only weekend | 2, 5, 11, 12, 13, 14, 15, 17, 18 |
| Pedestrian all year round | 16 |

*5.3. Phase 3. Proposals*

As a result of Phase 3 Proposals, an example of one of the urban regeneration proposals for the neighbourhood is shown in Figure 9.

Each planned project action is structured in a seven-step timetable:

- Site inspection;
- Dialogue between citizens and associations;
- Drafting of guidelines and intervention;
- Presentation of the Intervention Guidelines to the Public Administration;
- Dialogue between public administration and private actors;
- Regeneration of degraded spaces;
- Realisation of temporary projects.

Specifically, the proposal described concerns the flexible design of some urban voids, for which the name 'Terraces in Bloom' was chosen.

The project plans for the regeneration of five buildings on Via dei Sabelli, indicated on the plan with red circles (15a–e). The axonometric render the project envisaged for lot 15c relates the state of the sites in two different moments of the project. The first refers to the setting up of the area for cultural events (implementing two of the temporary functions made explicit in the general diagram: "music and singing", "tables and refreshments"); the second, for commemoration days (implementing of one of the temporary functions made explicit in the general diagram; namely, "music and singing").

In addition, a 'calendar of events' is also indicated, from January to December, defining the functionality of the space throughout the year. From this, it can be deduced that the area will predominantly host cultural events and, therefore, there will be a set-up specifically dedicated to these activities. However, on some specific days, the area will be dedicated to celebrations or commemorations, and in these cases there will be a set-up specifically dedicated to these activities.

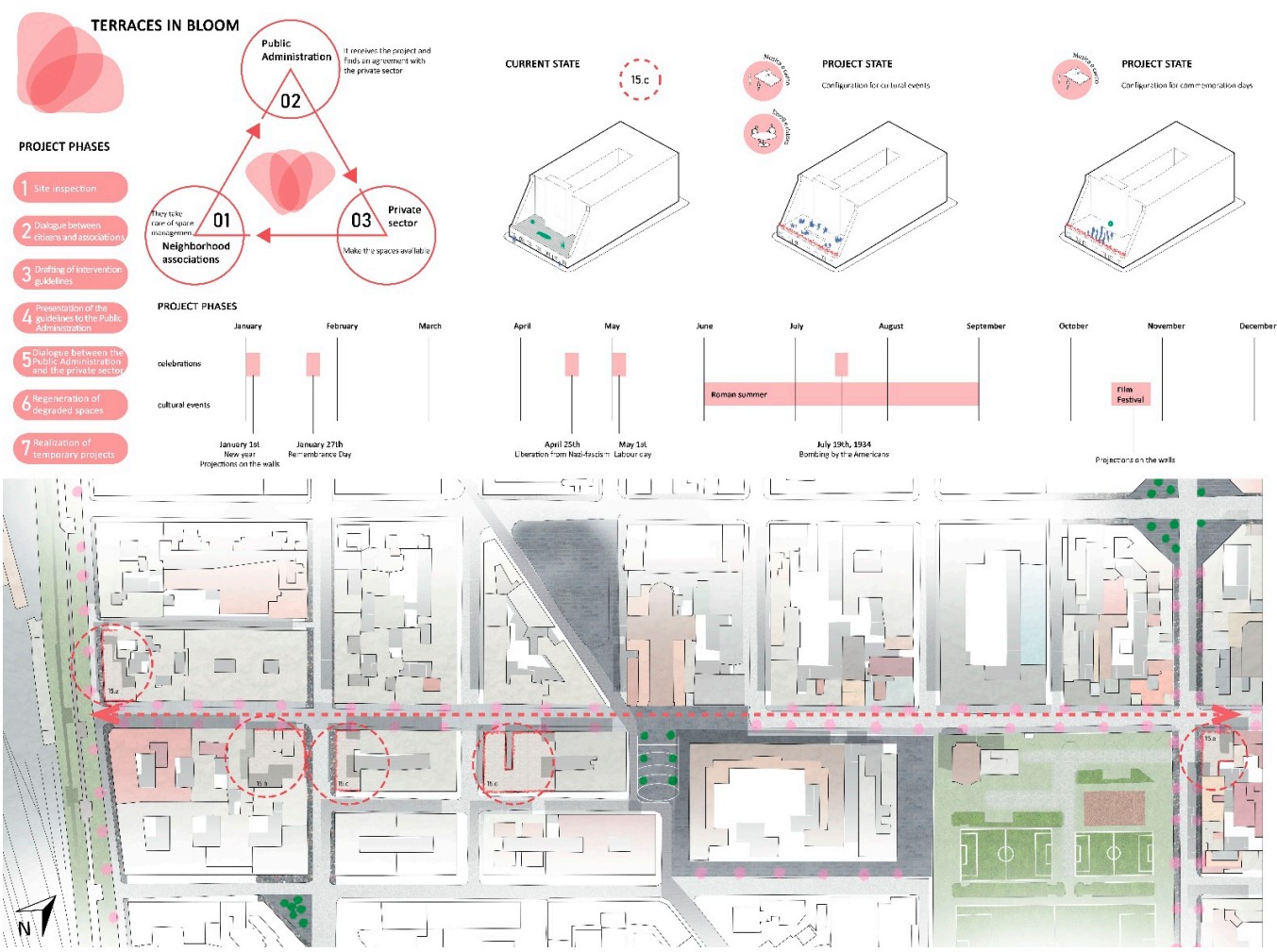

**Figure 9.** Schematic example of project hypothesis for open spaces in the San Lorenzo district (original scale 1:500).

## 6. Conclusions

The experimentation carried out in the case study represented the place where the grid of theoretical–methodological and operational references identified, as a result of examination of the main bibliographic references, current research and emblematic best practices converged. This was verified and actualized; in its entirety it constituted the methodological reference of this research activity.

The results of these activities highlight the presence of some recurring themes that constitute possible theoretical–methodological and operational references, replicable and exportable in other territorial contexts. They are about the specificity of the places around which the possible success and quality of the interventions revolve:

- the theme of the inter-scalarity of the project, and the possibility of declining and experimenting with such operations at the territorial, local and micro-scale, through structured planning of interventions and the gradualness of project actions;
- the relationship and integration with the physical context of the existing urban structure, with particular attention to aspects of connectivity, usability, user-friendliness, livability, safety of spaces, aspects of ecological-environmental sustainability, and landscape components;
- the flexibility of uses and functions, which also implies modularity and replicability of certain solutions that allow, at the same time, the recognizability of a unified project;

- innovation in the identification of area acquisition mechanisms, as in the case of the *Estonoesunsolar Programme*'s land occupation plan, which provides for a temporary transfer of areas by private individuals and a simultaneous appropriation by local communities;
- the involvement of the local community in co-working and co-production activities in open and neighbourhood spaces. Consequently, thehe realization of tangible and usable common goods, with spin-offs in terms of social inclusion and socio-economic development;
- the possible involvement of multidisciplinary expertise for aspects related to new technologies, environmental sustainability issues, social and economic impacts of interventions, spatial design, etc.;
- the return to a central role for the public entity as a promoter of interventions, the involvement of local communities in the governance of public space, management, and participation procedures.

These methodological and operational references will be able to innovate the planning and design policies of public spaces through a regulation of urban uses and spaces that acts on the dynamics of space and time. This is consistent with the theme of proximity as a factor on which to calibrate the spatial reorganization of services, functions, and the management of social dynamics [79].

Starting from this change of paradigm, it is possible to imagine that among the orientations of the territorial government bodies in the use of European funds (Next Generation EU, Horizon Europe, ERDF, New European Bauhaus), an increasing space will be reserved for experimental measures and pilot actions that redevelop spaces. This may be through forms of tactical urbanism and paths of active citizen participation. A scenario in which major infrastructural interventions can be rethought and integrated with the activation of creative-innovation experiences with a strong social impact, tested in a pilot form on a local and/or district scale.

In this sense, as already emphasized, the role of public government in the promotion and procurement of financial resources, as well as in the coordination of the different stages of the process and the concertation between the various stakeholders involved in the implementation of the project, appears to be central.

A further element of complexity in the implementation of the proposed methodology is the measurement, using indicators and parameters, of the results from the point of view of the social and economic impacts on the territory, and the short- and long-term monitoring of the quality of the interventions.

**Author Contributions:** Conceptualization, C.M.; methodology, C.M., I.G.L., P.D.M.; software, C.M.; validation, C.M., I.G.L., P.D.M.; format analysis, C.M., I.G.L., P.D.M.; investigation, C.M., I.G.L., P.D.M.; resources, C.M.; data curation, C.M.; writing-original draft preparation, C.M., I.G.L., P.D.M.; writing-review and editing, C.M.; supervision, C.M.; project administration, C.M.; funding acquisition, C.M. All authors have read and agreed to the published version of the manuscript.

**Funding:** This research received no external funding.

**Institutional Review Board Statement:** Not applicable.

**Informed Consent Statement:** Not applicable.

**Data Availability Statement:** Not applicable.

**Acknowledgments:** The paper is the result of a shared reflection by the authors. However, paragraphs 1–2–4–5 are to be attributed to Carmela Mariano; paragraph 3.1 is to be attributed to Ignacio Gràvalos Lacambra and Patrizia Di Monte; paragraphs 3.2 and 6 are to be attributed to all three authors. The research and experimentation activity took place in Rome in 2021, when Ignacio Gravalos Lacambra in Architectural design, Escuela de Arquitectura, Universidad San Jorge, Zaragoza, Department of Planning, Design, Technology of Architecture (scientific head Carmela Mariano). The research activity also saw the participation of Patrizia Di Monte Technical Office Urban Regeneration

Estonoesunsolar. Students from the master's degree Course in Architecture-Urban Regeneration were also involved in the activities.

**Conflicts of Interest:** The authors declare no conflict of interest.

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
