# Peer review of "Open Urban Space Regeneration Strategies Based on Urban Welfare: A Project and Experiment in the San Lorenzo District in Rome, Italy"

_sustainability, doi:10.3390/su142416487_

Round 1
Reviewer 1 Report
This is an interesting study and very well organized. I have a few minor remarks that the authors should address before publication:
1. English check is required to eliminate grammatical errors as well as the structure of the sentences (includes quite of few long sentences which could be difficult for readers to understand).
2. Introduction and Literature review sections are written very well, however, I missed the discussion on the main contribution and urgency of the study.
3. The authors could present some of the results in Tables making them easier to follow.
4. Some discussion on policy implications as well as the main limitations of the study is required.
Author Response
"Please see the attachment."

Reviewer 2 Report
Dear authors,
The research is quite interesting and it can be seen that the whole project was fully finished. But in the manuscript, many aspects should still be revised. Especially, how the research methodology is proposed based on previous research, this part should be fully explained. The writing structure should be clearer. More suggestions are as follows:
1. The title should be revised, here is a suggestion:
Open Urban Space Regeneration Based on Urban Welfare: A Project and Experiment in San Lorenzo district in Rome, Italy
2. Abstract should start with the research background, not properly the research methodology. The research results obtained should also be explained in the abstract.
3. Introduction is too general. Can you provide also some examples and data/number about Covid influences on urban design or urban life.
4. The paragraphs are too short; you can combine two short ones into one long paragraph.
5. After literature review, it’s not clear how your research methodology was developed, mainly based on which research? And in the literature review, more discussions on urban regeneration and urban Welfare should be added.
6. in "4 Case study", the site of San Lorenzo should be added with a map (its location in Rome and Italy). The case study is too long, can you divide it into 2-3 subsections?
7. The “pilot experience” can move to a new subsection “3.2 pilot experience”.
8. In your case study, for the three phrases “offer, Evaluation and Demand, Proposals”, it will be better to subdivide into several subsections.
Thank you.
Author Response
"Please see the attachment."

Reviewer 3 Report
Introduction represents a good overview of the topic, but without clear aims and without explaining what the need for conducting such study is. This should be elaborated in a more straight-to-the-point way. So, the reader doesn’t realize in the beginning why is study conducted.
The study has a high value, but the authors should explicitly explain how this contributes to the sustainability of the city with some examples.
The study is highly practical, so it would be good if the authors could state some possible theoretical contributions as well (in conclusion part)
The conclusion should mention the limitations of the study but also some plans for future research. At the moment it looks like very unfinished.
Author Response
"Please see the attachment."

Round 2
Reviewer 2 Report
Dear authors,
It can be seen the revisions are made carefully. Below are a few other suggestions:
1. The title should not be two sentences, it should be: Open Urban Space Regeneration Strategies Based on Urban Welfare: A Project and Experiment in the San Lorenzo District in Rome, Italy.
2. The abstract should still be revised. First, introduce the research background, not directly introduce your methodology.
3. The figures are not very scientifically qualified, especially the maps and design works (from figure 4-9), north arrow and scales should strictly be added to the map.
Reviewer 3 Report
We would like to thank to the authors for considering all the comments. The paper is now improved and I suggest acceptance.
